# Strategies to Improve Cannabidiol Bioavailability and Drug Delivery

**DOI:** 10.3390/ph17020244

**Published:** 2024-02-13

**Authors:** Saoirse Elizabeth O’Sullivan, Sanne Skov Jensen, Aditya Reddy Kolli, Gitte Nykjær Nikolajsen, Heidi Ziegler Bruun, Julia Hoeng

**Affiliations:** 1CanPharmaConsulting, Nottingham NG9 3BB, UK; 2Fertin Pharma, Dandyvej 19, 7100 Vejle, Denmark; sanj@fertin.com (S.S.J.); nykj@fertin.com (G.N.N.); hzp@fertin.com (H.Z.B.); 3PMI R&D, Philip Morris Products S.A., Quai Jeanrenaud 5, 2000 Neuchâtel, Switzerland; adityareddy.kolli@pmi.com; 4Vectura Fertin Pharma, 4058 Basel, Switzerland; julia.hoeng@vecturafertinpharma.com

**Keywords:** cannabidiol, clinical, pharmacokinetics, pharmacodynamics, disease, route of administration, tissue distribution

## Abstract

The poor physicochemical properties of cannabidiol (CBD) hamper its clinical development. The aim of this review was to examine the literature to identify novel oral products and delivery strategies for CBD, while assessing their clinical implications and translatability. Evaluation of the published literature revealed that oral CBD strategies are primarily focused on lipid-based and emulsion solutions or encapsulations, which improve the overall pharmacokinetics (PK) of CBD. Some emulsion formulations demonstrate more rapid systemic delivery. Variability in the PK effects of different oral CBD products is apparent across species. Several novel administration routes exist for CBD delivery that may offer promise for specific indications. For example, intranasal administration and inhalation allow quick delivery of CBD to the plasma and the brain, whereas transdermal and transmucosal administration routes deliver CBD systemically more slowly. There are limited but promising data on novel delivery routes such as intramuscular and subcutaneous. Very limited data show that CBD is generally well distributed across tissues and that some CBD products enable increased delivery of CBD to different brain regions. However, evidence is limited regarding whether changes in CBD PK profiles and tissue distribution equate to superior therapeutic efficacy across indications and whether specific CBD products might be suited to particular indications.

## 1. Introduction

The clinical use of cannabidiol (CBD) is receiving considerable attention, with clinical evidence supporting its efficacy in multiple disorders, including anxiety, psychosis, and schizophrenia, among others [1]. Epidiolex^®^ is currently the only US Food and Drug Administration (FDA)-approved purified CBD product, which is licensed for the treatment of childhood intractable epilepsy. Epidiolex is an oral solution, with an estimated oral bioavailability of 6–14% [2]. The mean effective dose of Epidiolex is between 20 and 25 mg/kg/day, corresponding to a mean plasma concentration of approximately 125 ng/mL [3,4]. Epidiolex is also being evaluated in the clinic for other paediatric diseases, including anxiety in epilepsy (NCT05324449), hereditary epidermolysis bullosa (NCT05651607), and autism spectrum disorder (NCT04517799), which may result in future label expansion.

The clinical development of CBD is somewhat hampered by its physicochemical properties, as described previously by Millar et al. [5]. As a lipophilic compound, CBD has poor solubility, variable pharmacokinetic (PK) profiles, poor oral bioavailability, and food interactions. To overcome the poor physicochemical properties of CBD, many investigators are developing novel strategies to improve oral bioavailability and identify novel delivery mechanisms for CBD. The primary aim of this review was to assess the preclinical and clinical PK of various CBD formulations, which have been developed using novel methodologies to enhance the bioavailability of CBD. The secondary aim was to review the PK of CBD administered by various routes for improved systemic exposure. Data from relevant studies were identified through a systematic literature search and identification of manuscripts investigating the PK profiles of purified CBD. Relevant manuscripts were also hand searched for any additional relevant studies. All authors independently searched for included studies. Figure 1 summarises the different routes of CBD administration across species. The data in Figure 1 showcase how different CBD products give rise to differences in maximum plasma concentration (C_max_), time to maximum plasma concentration (T_max_), and total drug exposure (area under the curve (AUC)). However, a head-to-head comparison of PK-based formulation performance across various routes of administration requires further derivation of PK indices. The two most important factors influencing in vivo formulation performance are the overall resulting systemic exposure and the rate or speed of systemic delivery. As different doses are administered across different routes, a dose-normalised AUC (AUC/dose) enables determination of systemic exposure per unit dose. While the rate of systemic delivery can be determined by C_max_/T_max_, a dose increase may result in a higher C_max_, yielding a higher C_max_/T_max_ for the same formulation. Therefore, dose-normalised rate of systemic delivery (C_max_/T_max_ /dose) allows for the comparison of different formulations administered at various doses.

## 2. Strategies to Improve the Oral Bioavailability of CBD (Preclinical Studies)

Oral CBD is patient-friendly and easy to consume; however, the poor oral bioavailability of CBD remains a challenge, warranting the development of CBD products for improved in vivo performance. A wide variety of novel oral formulations are evaluated in preclinical animal models across multiple species. Here, we review the performance of different oral formulations of CBD that have been designed to improve the PK of CBD. A comparison of preclinical PK data based on the AUC/dose and C_max_/T_max_/dose of these oral formulations is shown in Figure 2, and the relevant data are listed in Table 1.

**Table 1 pharmaceuticals-17-00244-t001:** PK parameters for oral administration of different CBD formulations in dogs, horses, mice, and rats. AUC, total drug exposure area under the curve; AUC/dose, dose-normalised area under the curve; CBD, cannabidiol; C_max_, maximum plasma concentration; C_max_/T_max_/dose, dose-normalised rate of systemic delivery; DCA, deoxycholic acid; EtOH, ethyl alcohol; LCT, long-chain triglyceride; MCT, medium-chain triglyceride; NP, nanoparticle; OG, oleoylglycerol; PG, propylene glycol; PK, pharmacokinetics; PNL, pro-nanoliposphere; SNEDDS, self-nanoemulsifying drug-delivery system; T_max_, time to maximum concentration.

Route	Dose (mg/kg)	Vehicle	Formulation Category	T_max_ (h)	C_max_ (ng/mL)	AUC (ng/mL × h)	C_max_ (ng/mL)/Dose (mg/kg)	AUC (ng/mL × h)/Dose (mg/kg)	C_max_ (ng/mL)/T_max_ (h)/Dose (mg/kg)	Ref
Rat										
Oral	19.3	MCT	Lipid-based	6	128	445	6.63	23.1	1.11	[6]
Oral	20	Sesame oil (Epidiolex)	Lipid-based	1	629	2766	31.45	138	31.5	[6]
Oral	12	Sesame oil (LCT)	Lipid-based	3	308	932	25.7	77.7	8.56	[7]
Oral	12	Sesame oil	Lipid-based	4	225	821	18.8	68.4	4.69	[8]
Oral	12	Oleic acid	Lipid-based	3	134	604	11.2	50.3	3.72	[8]
Oral	12	Linoleic acid	Lipid-based	4	108	602	9.00	50.2	2.25	[8]
Oral	12	2-OG + oleic acid	Lipid-based	6	71	512	5.92	42.7	0.99	[8]
Oral	12	Oleic acid + glycerol	Lipid-based	3	125	584	10.4	48.7	3.47	[8]
Oral	12	Glycerol trioleate	Lipid-based	3	154	560	12.8	46.7	4.28	[8]
Oral	25	Sesame oil	Lipid-based	2.5	724	2702	29.0	108	11.6	[9]
Oral	25	Sesame oil + surf	Lipid-based	2	562	2131	22.5	85.2	11.2	[9]
Oral	12	Sesame oil	Lipid-based	3	164	865	13.7	72.1	4.56	[10]
Oral	12	Coconut oil	Lipid-based	5	84	413	7	34.4	1.40	[10]
Oral	12	Rapeseed oil	Lipid-based	5	118	587	9.83	48.9	1.97	[10]
Oral	10	Sunflower oil	Lipid-based	2	96.5	292	9.65	9.30	1.40	[11]
Oral	12	Sesame oil	Lipid-based	3	209	865	17.4	72.1	5.81	[12]
Oral	12	Soybean oil	Lipid-based	3	165	775	13.8	64.6	4.58	[12]
Oral	12	Peanut oil	Lipid-based	3	153	737	12.8	61.4	4.25	[12]
Oral	12	Olive oil	Lipid-based	2	258	835	21.5	69.6	10.75	[12]
Oral	12	Sunflower oil	Lipid-based	3	112	551	9.33	45.9	3.11	[12]
Oral	12	Coconut oil	Lipid-based	5	96	413	8	34.4	1.60	[12]
Oral	100	Olive oil	Lipid-based	8	2720	26,700	27.2	267	3.40	[13]
Oral	15	Sesame oil	Lipid-based	4	136	660	9.07	44.0	2.27	[14]
Oral	18.75	Sunflower oil	Lipid-based	4.05	8.95	66.8	0.48	3.56	0.12	[15]
Oral	20	Nanoemulsion	Emulsion-based	1	454	1949	22.7	97.5	22.7	[6]
Oral	18.75	Sunflower oil + lecithin	Emulsion-based	2.54	34.4	153	1.83	8.16	0.72	[15]
Oral	15	Sesame-SNEDDS	Emulsion-based	1.08	137	611	9.13	40.7	8.46	[16]
Oral	15	MCT-SNEDDS	Emulsion-based	1	101	579	6.73	38.6	6.73	[16]
Oral	15	Cocoa butter-SNEDDS	Emulsion-based	6	458	2864	30.5	191	5.09	[16]
Oral	15	Tricaprin-SNEDDS	Emulsion-based	5	261	2041	17.4	136	3.48	[16]
Oral	15	Piperine-PNL	Emulsion-based	2	178	593	11.9	39.5	5.93	[16]
Oral	15	Piperine-PNL	Emulsion-based	1.8	168	809	11.2	53.9	6.22	[16]
Oral	15	PNL	Emulsion-based	1.2	130	286	8.67	19.1	7.22	[16]
Oral	15	PNL	Emulsion-based	1.11	137	300	9.13	20.0	8.23	[17]
Oral	15	Piperine-PNL	Emulsion-based	1.67	170	570	11.3	38.0	6.79	[17]
Oral	15	Curcumin-PNL	Emulsion-based	1.56	63	168	4.2	11.2	2.69	[17]
Oral	15	Resveratrol-PNL	Emulsion-based	1.9	96	202	6.4	13.5	3.37	[17]
Oral	50	Nanoemulsion	Emulsion-based	2.4	3230	22,400	64.6	448	26.9	[13]
Oral	40	MCT protein–maltodextrin	Emulsion-based	4	1190	8560	29.8	214	7.44	[18]
Oral	40	MCT/LCT protein–maltodextrin	Emulsion-based	2	2120	12,190	53	305	26.5	[18]
Oral	40	LCT protein–maltodextrin	Emulsion-based	2	1890	12,500	47.3	313	23.6	[18]
Oral	14.5	Pure CBD suspension	Liquid	4	1.1	3.7	0.08	0.26	0.02	[19]
Oral	10	PG	Liquid	2	28	93	2.80	147	32.4	[20]
Oral	12	PG	Liquid	2	81	356	6.75	29.7	3.38	[12]
Oral	12	PG	Liquid	3	87	327	7.25	27.3	2.42	[7]
Oral	12	PG:EtOH:water	Liquid	5	55	356	4.58	29.7	0.92	[10]
Oral	40	Zein NP whey protein	Nanotechnology	2	466	2912	11.65	72.8	5.83	[21]
Oral	12.6	Nanocrystals	Nanotechnology	1.25	151	847	12.0	67.2	9.61	[14]
Oral	14.5	Nanoparticles	Nanotechnology	0.3	21	51.9	1.45	3.58	4.83	[19]
Oral	40	Free-form CBD	Solid/Resin	4	232	1657	5.80	41.4	1.45	[21]
Oral	15	Lipid-free	Solid/Resin	1.07	39	90	2.60	6.00	2.43	[17]
Oral	48.00	Isolate	Solid/Resin	3	739	5307	15.4	111	5.13	[22]
Horse										
Oral	10	Sesame oil	Lipid-based	3.5	55.7	778	5.57	77.8	1.59	[23]
Oral	10	Micellar	Emulsion-based	2	143	830	14.3	83.0	7.14	[23]
Dog										
Oral	5.77	Oil	Lipid-based	2	625	147	108	25.5	54.2	[24]
Oral	5.77	Microencapsulated oil beads	Lipid-based	6	346	104	60.0	17.9	10	[24]
Oral	11.5	Oil	Lipid-based	4	846	317	73.6	27.5	18.3	[24]
Oral	11.5	Microencapsulated oil beads	Lipid-based	4	578	177	50.3	15.4	12.5	[24]
Oral	1	MCT	Lipid-based	2.17	207	648	207	648	95.3	[25]
Oral	8.33	Cannef tablets	Solid/Resin	3.5	217	1376	26.1	165	7.43	[26]
Transmucosal	1	MCT	Lipid-based	1.92	200	536	200	536	104	[25]
Mice										
Oral	5	MCT (Miglyol 812 N)	Lipid-based	0.3	8	14.9	1.60	2.98	5.13	[27]
Oral	5	Capsules	Emulsion-based	0.3	10.9	16.3	2.18	3.26	7.27	[27]
Oral	5	Capsules + DCA	Emulsion-based	1	11	6.5	2.20	1.30	2.2	[27]
Oral	20	Soybean oil + fat	Emulsion-based	2	130	551	6.48	27.6	3.24	[28]
Oral	30	Gel	Solid/Resin	1	236.2	428	7.87	14.3	7.87	[29]

CBD is lipid-soluble; therefore, many studies have investigated the PK profiles of lipid formulations of CBD, such as CBD formulated with coconut, olive, peanut, rapeseed, sesame, soybean, and sunflower oil (Table 1). In rats, lipid-based CBD formulations result in a wide range of systemic exposure levels according to the AUC/dose, ranging from 3.6 to 267 (Table 1). For example, one study showed that compared with a lipid-free formulation, a sesame oil-based CBD formulation resulted in a 14% increase in absolute bioavailability, a higher C_max_, 3-fold higher systemic exposure (AUC), and no alterations in T_max_ in rats [7]. CBD as part of a sesame-oil-based lipid formulation appears to be rapidly absorbed into the systemic circulation in rats, with a maximal C_max_/T_max_/dose of 31.5 and an AUC/dose of 138 [6]. Lipid formulations, such as CBD formulated with sesame oil, also lead to higher lymphatic uptake of CBD with a lymphatic-to-plasma ratio of 250:1 [30]. Delivery of sesame-oil-based and olive-oil-based formulations increases AUC/dose, with rapid systemic absorption, as indicated by a higher C_max_/T_max_/dose, in rats. Other oils, such as soybean, peanut, sunflower, rapeseed, and coconut oil, show lower AUC/dose and C_max_/T_max_/dose values (Table 1). Most studies have shown that sesame-oil-based CBD formulations have superior PK performance in rats, while medium-chain triglyceride (MCT)-based formulations are most rapidly absorbed in dogs with maximal systemic exposure. However, it must be noted that variability exists among studies, and the number of studies is limited. As such, more research is required (Figure 2).

The PK parameters of emulsion-based CBD formulations have been evaluated in 18 preclinical studies (Table 1). Depending on the emulsifying technology and formulation, the AUC/dose in rodents ranges between 8.2 and 448. Jelinek and colleagues showed that a lecithin-based microemulsion increased CBD bioavailability, with a 3.2-fold higher C_max_ and a 2.1-fold higher AUC than the sunflower-oil-based formulation [15]. The CBD concentration in the lymph was 2–3 orders of magnitude higher than the serum concentration for both the lecithin-based formulation and the sunflower-oil-based formulation, emphasising their absorption potential. CBD emulsions with whey protein–maltodextrin are among the most efficient CBD formulations tested to date [18]. The addition of MCTs or long-chain triglycerides (LCTs) improves the delivery efficiency of CBD (Figure 2), and in vivo MCT/LCT and LCT oils show better CBD transport and absorption than lipid-free formulations.

Nanoemulsions demonstrate several advantages, such as a small droplet size, large surface area, and high solubilisation capacity, resulting in good in vivo performance with a C_max_/T_max_/dose of 26.9 and an AUC/dose of 448 [13]. Self-nanoemulsifying drug-delivery systems (SNEDDSs) are anhydrous homogeneous liquid mixtures consisting of oil, surfactant, active pharmaceutical ingredient, and co-emulsifier or solubiliser, forming an oil-in-water nanoemulsion (≤200 nm) upon dilution with water. Izgelov and colleagues [16] compared the PK profiles of LCT-based and MCT-based CBD formulations in rats to understand if incorporating LCTs into SNEDDSs would enhance CBD delivery owing to lymphatic targeting. The results showed that incorporation of LCTs into SNEDDSs enhanced CBD bioavailability when compared with MCT oil; however, this may depend on the triglyceride levels. In vivo evaluation of cocoa butter (LCT-SNEDDS) and tricaprin (MCT-SNEDDS) showed that the LCT-SNEDDS had a 1.6-fold higher AUC than the MCT-SNEDDS. While MCT-based and sesame-oil-based SNEDDS formulations were rapidly absorbed (C_max_/T_max_/dose), cocoa butter-based and tricaprin-based SNEDDS formulations achieved higher systemic exposure (AUC/dose) (Figure 1). Cherniakov and colleagues [17] developed pro-nanoliposphere (PNL) SNEDDSs composed of lipid and emulsifying excipients known to increase solubility, reduce phase I metabolism, and improve absorption of lipophilic active compounds using absorption enhancers, such as piperine. Compared with CBD in solution (propylene glycol:ethyl alcohol:water), CBD–PNL–piperine showed a six-fold increase in AUC [17]. However, another study reported no difference in the enhancement of CBD bioavailability by piperine when administered chronically or acutely in conjunction with the PNL SNEDDS in rats [16]. In fact, the cocoa-butter-based and tricaprin-based SNEDDS formulations showed superior performance to the PNL-based formulation. Kok and colleagues [6] compared the PK profiles of SNEDDS formulations with the MCT-CBD and sesame-oil-based CBD formulations in healthy female rats. No PK differences were found between the two SNEDDS formulations; however, the SNEDDSs showed 2.2- and 2.8-fold greater systemic exposure than MCT-CBD, while there was no increase in systemic exposure when compared with the sesame-oil-based CBD formulation.

Shreiber-Livne et al. [19] synthesised CBD-loaded poly(ethyleneglycol) (PEG) b poly(epsilon caprolactone) nanoparticles using a nanoprecipitation method. Compared with CBD in suspension, the CBD-loaded nanoparticles showed a 14-fold increase in AUC and a reduction in T_max_. Wang et al. [21] found that CBD formulated in a nanoparticle composite of zinc and whey protein demonstrated increased water solubility and a 2-fold and 1.75-fold increase in C_max_ and AUC, respectively, compared with a soybean-oil-based formulation. However, the overall performance was similar to that of the soybean-oil-based formulation.

In summary, the strategies used to increase the oral bioavailability of CBD in the preclinical space are predominantly lipid-based formulations. Incomplete gastrointestinal absorption combined with extensive hepatic metabolism are the main reasons for the low bioavailability of CBD. The putative mechanism of increased CBD absorption in the presence of lipids is enhanced intestinal–lymphatic transport, minimising presystemic elimination and hepatic first-pass metabolism. In general, emulsion-based formulations and lipid-based formulations demonstrate better preclinical PK performance by taking advantage of the physicochemical properties of CBD, such as its lipophilicity, to increase bioavailability. Whereas SNEDDSs often improve the speed of delivery and demonstrate higher C_max_ values, lipid-based formulations prolong the plasma concentration of CBD. Further optimisation of parameters affecting the gastrointestinal absorption and presystemic metabolism of CBD could facilitate the development of novel formulations that demonstrate better and more consistent bioavailability, as well as reduced variability in the clinic. Alternative technologies are continuously being developed and patented to increase CBD solubility and bioavailability. Some examples include the single CBD crystal structure (Pureform), co-crystals (Artelo Biosciences, CA, US [31]), CBD and cyclodextrin complexes (Medexus Pharmaceuticals and Vireo Health LLC, Canada [32]), encapsulation (Avecho, Australia [33]), and CBD encapsulated in biodegradable polymer nanospheres as a lyophilised powder (Aphios Corporation, MA, US [34]). These formulations are often combined with technologies that are intended to increase the delivery of CBD and cannabinoids from rapidly disintegrating tablets [35], lozenges [36], and liquifying tablets (Fertin Pharma) [37].

The translation of formulation performance across species is challenging. The overall in vivo exposure (AUC/dose) to CBD is low in mice, despite a comparable rate of systemic delivery (C_max_/T_max_/dose) to humans and rats (see Figure 3). This may be due to differential tissue distribution kinetics and higher systemic clearance; however, the exact mechanisms need to be further evaluated. Whereas dogs have a comparable AUC/dose values to humans and rats, the rate of systemic delivery tends to be higher (C_max_/T_max_/dose) in dogs. Emulsion-based formulations have shown improved performance compared with lipid-based formulations in rats, whereas oral administration of lipid-based formulations has shown superior performance in dogs. It is because of potential differences in absorption, distribution, metabolism, and excretion that CBD formulations have commonly been evaluated in rats. This is because rats tend to show comparable AUC/dose and C_max_/T_max_/dose values to humans, making rats a relevant animal model for the preclinical evaluation of CBD products.

## 3. Strategies to Improve the Oral Bioavailability of CBD (Clinical Studies)

Clinical evaluation of novel CBD formulations is primarily focused on lipid- and emulsion-based delivery systems. A comparison of the clinical PK profiles of CBD formulations based on the AUC/dose and C_max_/T_max_/dose values of different oral formulations is shown in Figure 4 and Table 2.

**Table 2 pharmaceuticals-17-00244-t002:** PK parameters for various CBD formulations administered by different routes in humans. AUC, total drug exposure area under the curve; AUC/dose, dose-normalised area under the curve; CBD, cannabidiol; C_max_, maximum plasma concentration; C_max_/T_max_/dose, dose-normalised rate of systemic delivery; EtOH, ethyl alcohol; GML, glyceryl monolinoleate; MCT, medium-chain triglyceride; NP, nanoparticle; PG, propylene glycol; PK, pharmacokinetics; PNL, pro-nanoliposphere; SEDDS, self-emulsifying drug-delivery system; SNEDDS, self-nanoemulsifying drug-delivery system; T_max_, time to maximum concentration.

Route	Dose (mg/kg) *	Vehicle	Formulation Category	T_max_ (h)	C_max_ (ng/mL)	AUC (ng/mL × h)	C_max_ (ng/mL)/Dose (mg/kg)	AUC (ng/mL × h)/Dose (mg/kg)	C_max_ (ng/mL)/T_max_ (h)/Dose (mg/kg)	Ref
Oral	0.33	MCT	Lipid-based	3	3.05	19.2	9.15	57.7	3.05	[38]
Oral	0.67	MCT	Lipid-based	5.2	14	73.8	21	111	4.04	[39]
Oral	0.13	MCT	Lipid-based	5.1	0.84	3.4	6.31	25.5	1.24	[40]
Oral	0.40	Lipid-soluble	Lipid-based	1.5	0.65	1.51	1.63	3.77	1.08	[41]
Oral	0.60	Generic	Lipid-based	1.88	16.8	37.5	28	62.6	14.9	[42]
Oral	1.20	Generic	Lipid-based	2.05	23.5	119	19.6	98.8	9.55	[42]
Oral	0.67	Epidiolex	Lipid-based	2.03	6.3	20.1	9.45	30.1	4.66	[43]
Oral	0.53	MCT	Lipid-based	4.26	3.5	11.9	6.55	22.4	1.54	[44]
Oral	0.53	GML	Lipid-based	1.59	6.47	15.8	12.1	29.5	7.6	[44]
Oral	1.20	Sesame oil (gelcap)	Lipid-based	4	14	66	11.7	55	2.92	[45]
Oral	1.33	Syrup	Lipid-based	3.2	2.8	10.3	2.1	7.73	0.66	[46]
Oral	1.33	Epidiolex	Lipid-based	3.3	20.5	64.1	15.4	48.1	4.66	[46]
Oral	1.6	MCT	Lipid-based	4.89	13.9	122	8.7	76.3	1.78	[47]
Oral	20	Epidiolex	Lipid-based	4	292	1517	14.6	75.9	3.66	[48]
Oral	40	Epidiolex	Lipid-based	5	533	2669	13.3	66.7	2.67	[48]
Oral	60	Epidiolex	Lipid-based	5	722	3215	12	53.6	2.41	[48]
Oral	80	Epidiolex	Lipid-based	5	782	3696	9.78	46.2	1.96	[48]
Oral	10	Epidiolex	Lipid-based	5.11	336	1587	33.6	159	6.58	[49]
Oral	20	Epidiolex	Lipid-based	6.13	525	2650	26.2	132	4.28	[49]
Oral	60	Epidiolex	Lipid-based	4.07	427	2339	7.12	39	1.75	[49]
Oral	5.33	Corn oil (gelcap)	Lipid-based	3	181	704	33.9	132	11.3	[50]
Oral	5.33	Corn oil (gelcap)	Lipid-based	1.5	114	482	21.4	90.4	14.3	[50]
Oral	10.7	Corn oil (gelcap)	Lipid-based	3	221	867	20.7	81.3	6.91	[50]
Oral	10.7	Corn oil (gelcap)	Lipid-based	4	157	722	14.7	67.7	3.68	[50]
Oral	2	Corn oil	Lipid-based	2	82.6	269	41.3	61.5	20.7	[51]
Oral	0.4	MCT	Lipid-based	1.28	3.5	13.81	8.85	34.5	6.91	[52]
Oral	0.4	Powder	Lipid-based	1.53	2.88	9.96	7.2	24.9	4.71	[52]
Oral	0.4	MCT	Lipid-based	0.7	5.57	10.77	13.9	26.9	19.9	[52]
Oral	0.33	SEDDS	Emulsion-based	1	13.5	32.6	40.6	97.9	40.6	[38]
Oral	0.13	Solutech	Emulsion-based	0.96	2	3.6	15	27	15.6	[40]
Oral	0.40	Water-soluble	Emulsion-based	0.9	2.82	6.8	7.05	17	7.83	[41]
Oral	0.53	SEDDS	Emulsion-based	1.68	6.94	17.7	13	33.3	7.74	[44]
Oral	1.20	SNEDDS (gelcap)	Emulsion-based	2	18	61	15	50.8	7.5	[45]
Oral	0.13	PNL	Emulsion-based	3	2.1	6.9	15.8	51.8	5.25	[53]
Oral	0.13	SEDDS	Emulsion-based	1.64	2.94	9.85	22.1	73.9	13.4	[54]
Oral	0.60	TurboCBD	Solidified	2.17	21.2	47.7	35.3	79.4	16.3	[42]
Oral	1.20	TurboCBD	Solidified	1.83	77.6	181	64.7	151	35.3	[42]
Oral	1.20	Powder (hardcap)	Solid/Capsule	8.4	0.8	8	0.67	6.67	0.08	[45]
Oral	1.33	Powder (gelcap)	Solid/Capsule	2.5	17.8	42.5	13.4	31.9	5.34	[46]
Oral	1.33	Gelcap	Solid/Capsule	4	11.1	31.5	8.33	23.6	2.08	[55]
Oral	0.13	Gelatin matrix pellets	Solid/Capsule	3	3.22	9.64	24.2	72.3	8.05	[56]
Oral	1.33	Gelatin matrix pellets	Solid/Capsule	3.5	47.4	150	35.6	112	10.2	[56]
Oral	10.7	Capsules	Solid/Capsule	3	77.9	580	7.30	54.4	2.43	[57]
Oral	0.13	Capsules	Solid/Capsule	1.27	2.47	6.03	18.5	45.3	14.6	[58]
Oral	2	Powder	Solid/Capsule	2.5	20.7	67.4	10.4	14.1	4.14	[51]
Transmucosal	0.08	Water-soluble NP	Emulsion-based	1	0.53	0.87	6.63	10.9	6.63	[59]
Transmucosal	0.24	Water-soluble NP	Emulsion-based	1	4.62	8.90	19.3	37.1	19.3	[59]
Transmucosal	0.13	Sativex	Liquid	3.5	2.05	7.3	15.4	54.8	4.39	[56]
Transmucosal	0.13	Sativex	Liquid	3.18	2.05	7.3	15.4	54.8	4.83	[54]
Transmucosal	0.13	Sativex	Liquid	1	0.5	3.1	3.75	23.3	3.75	[53]
Transmucosal	0.27	Sativex	Liquid	4.5	4.6	29.3	17.3	110	3.83	[39]
Transmucosal	0.13	EtOH:PG	Liquid	1.63	2.5	7.12	18.8	53.4	11.5	[58]
Transmucosal	0.13	EtOH:PG	Liquid	2.8	3.02	6.8	22.7	51	8.09	[58]
Transmucosal	0.13	EtOH:PG	Liquid	1.39	1.15	5.64	8.63	42.3	6.21	[60]
Transmucosal	0.13	EtOH:PG	Liquid	4	3.66	23.1	27.5	173	6.86	[60]
Transmucosal	0.13	EtOH:PG	Liquid	1	1.03	5.1	7.73	38.3	7.73	[61]
Transmucosal	0.13	EtOH:PG	Liquid	1.38	0.66	3.54	4.95	26.6	3.59	[61]
Transmucosal	0.13	EtOH:PG	Liquid	1.15	0.63	3	4.73	22.5	4.11	[61]
Transmucosal	0.07	Sativex	Liquid	3.7	1.6	4.5	24	67.5	6.49	[62]
Transmucosal	0.20	Sativex	Liquid	4	6.7	18.1	33.5	90.5	8.38	[62]
Transmucosal	0.33	Wafer	Solidified	4.5	9.1	33.5	27.3	101	6.07	[39]
Transmucosal	0.67	Wafer	Solidified	4.1	15	71	22.5	107	5.49	[39]

* Dose was normalised to a 75 kg human.

### 3.1. Lipid-Based Formulations

The PK profiles of CBD in lipid-based formulations (up to 80 mg/kg/day) have been evaluated in multiple clinical studies. Dramatic variations in the rate of systemic delivery and overall exposure have been observed for lipid-based CBD formulations in humans. MacNair and colleagues investigated potential sex differences in CBD PK profiles and found no differences in standard PK parameters [47]. Low and high doses of CBD in a sesame-oil-based formulation (Epidiolex) resulted in lower AUC/dose values, whereas an Epidiolex dose of approximately 10 mg/kg showed an increased AUC/dose value. Compared with rodents (or other preclinical species), higher doses of Epidiolex in humans resulted in lower AUC/dose and C_max_/T_max_/dose values, which may be attributed to potential saturation of CBD solubility in the gastrointestinal tract [63]. Although corn-oil-based CBD formulations have not been evaluated in preclinical species, these formulations [50,51] have demonstrated comparable systemic bioavailability (AUC/dose) to sesame-oil-based formulations (Epidiolex) in humans [43,46,48,49], with higher rates of systemic delivery. In one study, CBD formulated with lipid-based syrup showed poor PK performance [46].

### 3.2. Emulsion-Based Formulations

Six clinical studies have evaluated the PK profiles of emulsion-based CBD formulations. Compared with lipid-based formulations, emulsion-based formulations have a higher rate of systemic delivery (Figure 3). Berl and colleagues [40] compared the PK profile of Solutech™-TC10, a liquid emulsion containing 23.4 mg cannabis oil (10.0 mg Δ-9-tetrahydrocannabinol (THC); 9.76 mg CBD), with MCT-based CBD oil in healthy adults. Solutech-TC10 showed a greater C_max_, greater elimination and absorption rate constants, and a faster T_max_ and shorter lag time and half-life for all analytes compared with MCT-based CBD oil. These results may be due to the smaller droplet size and larger surface area of the nanoemulsion formulation. Another study compared the PK profile of CBD delivered with either MCT oil, glyceryl monolinoleate (GML), or as part of a self-emulsifying drug-delivery system (SEDDS) [44]. In vitro digestion showed that the SEDDS yielded the highest CBD recovery in the aqueous phase (86% ± 2%), followed by GML (13% ± 2%) and MCT oil (5.6% ± 0.8%), respectively. Clinical testing of the SEDDS confirmed better performance in humans, with the CBD AUC0–12 h being 1.12-fold and 1.48-fold higher than with the GML and MCT oil formulations, respectively. The use of proprietary VESIsorb^®^ SEDDS technology improved CBD bioavailability compared with the MCT–coconut oil CBD formulation [38]. The SEDDS formulation was superior for all measured PK parameters, including C_max_ and AUC0–24 h, as well as demonstrating a faster T_max_.

Several clinical investigations have demonstrated enhanced systemic delivery of CBD when using SEDDSs. Cherniakov and colleagues [53] compared the PK profile of a capsule-based THC-CBD-piperine-PNL delivery system with an equivalent oromucosal dose of Sativex^®^. The PNL formulation showed a 4-fold increase in C_max_ and a 2.2-fold increase in AUC compared with Sativex. Atsmon and colleagues [54] compared the proprietary PTL401 SEDDS, an oral THC and CBD formulation, with Sativex. The relative CBD bioavailability of PTL401 was 131%, with a 1.6-fold higher C_max_ when compared with the equivalent dose of Sativex. PTL401′s CBD T_max_ of 1.3 h was significantly shorter than Sativex’s^®^ T_max_ of 3.5 h.

Izgelov and colleagues [45] compared the oral absorption of synthetic CBD in a capsule, CBD dissolved in sesame oil, and CBD in a capsule-based SNEDDS. The administration of CBD in sesame oil resulted in a 17-fold increase in C_max_ and 8-fold AUC improvement when compared with CBD in powder form. This is in line with findings showing that CBD administered in corn oil produced a significantly higher plasma CBD concentration than pure powdered CBD [51]. The SNEDDS-based CBD formulation showed a 22-fold higher C_max_ and a 7-fold higher AUC than the powdered form [45]; however, when comparing the SNEDDS-based CBD formulation with the sesame-oil-based CBD formulation, there was no difference between the AUC and C_max_ values. T_max_ was delayed (median 4 h) with the sesame-oil-based and powdered formulations. Sesame-oil-based CBD showed delayed absorption, whereas the SNEDDS formulation resulted in a uniform early absorption profile. In other studies, a water-soluble emulsion-based formulation resulted in good systemic delivery but poor dose-normalised exposure [41,59].

### 3.3. Solidified Formulations

Hobbs and colleagues [41] compared the PK profile of a water-soluble CBD formulation with an MCT oil-based CBD formulation. The water-soluble powdered and MCT formulations demonstrated C_max_ values of 2.82 ng/mL and 0.65 ng/mL and T_max_ values of 54 min and 90 min, respectively. Interindividual variability was pronounced in the water-soluble CBD group, with some individuals showing a single absorption peak and others showing two peaks. This result is in accordance with the findings of sesame-oil-based CBD formulations, suggesting that (second) exposure may occur after potential enterohepatic recirculation [45]. The volume of CBD distribution was high for both treatment groups (water-soluble CBD and MCT-oil-based CBD) [41], suggesting that a large amount of CBD was partitioned to the tissues relative to the plasma. Williams and colleagues [52] compared the PK performance of five different CBD preparations diluted in water: a tincture, two powdered formulations, and two liquid formulations with varying combinations of MCT oil, emulsifiers, and polyols. One of the liquid formulations, which combined gum arabic, MCT oil, and citric acid, outperformed pure CBD powder and the CBD tincture according to the C_max_ and AUC0–4 h values. Another study evaluated the dose-dependent plasma exposure to CBD using the capsule-based TurboCBD™ delivery technology [42]. TurboCBD relies on dehydration of long-chain fatty acids high in oleic acid, as well as the addition of American ginseng, Ginkgo biloba, and CBD in the ratio of 13:3:1. Although there were no differences between TurboCBD and organic multi-spectrum hemp oil in the 45 mg condition, the circulating CBD concentration was higher in the TurboCBD 90 mg group at both 90 min (+86%) and 120 min (+65%) than in the 90 mg hemp oil control group. Only the TurboCBD 90 mg dose showed greater CBD bioavailability than placebo at 30 min, which remained elevated at 4 h.

### 3.4. Encapsulated Formulations

Atsmon and colleagues [56] compared the PK profile of proprietary agent PTL101 (containing 10 mg CBD), which is an oral gelatin matrix pellet containing CBD embedded in seamless gelatin matrix beadlets, with that of an equivalent dose of Sativex. The bioavailability of CBD with the 10 mg PTL101 dose was 131% relative to the corresponding dose from Sativex. PTL101 administration led to a 1.7-fold higher C_max_ and a 1.3-fold higher AUC when compared with Sativex. The T_max_ of both formulations occurred at approximately 3–3.5 h after administration; however, there was a 1 h lag time in CBD absorption when delivered via PTL101. Other studies have evaluated the exposure and elimination kinetics of CBD after oral or vaporised administration by measuring plasma [46] and urine [64] CBD concentrations over time. For oral administration, 100 mg CBD was delivered in a gelcap filled with inert microcrystalline cellulose, a CBD powder suspended in 2 mL ORA-Plus^®^, or as Epidiolex. ORA-Plus is a blend of suspending agents that have a high degree of colloidal activity, forming a gel-like matrix, which suspends particles and attenuates settling. Of the three oral formulations, Epidiolex had the highest C_max_, and the ORA-Plus syrup containing CBD had the lowest C_max_. All formulations demonstrated high variability. Epidiolex led to a higher urinary CBD concentration than the other oral formulations [64]. The T_max_ of all oral formulations was approximately 4–5 h [64]. Compared with other encapsulated formulations, such as gel capsules and powder, gelatin matrix pellets demonstrate faster systemic delivery and higher systemic exposure (Figure 4).

To summarise the human data on CBD oral bioavailability, the improvements of T_max_ and systemic delivery are commonly shown in published, clinically evaluated SEDDS and lipid-based formulations. Commercial products including technologies such as the VESIsorb^®^, TurboCBD, and PTL101 can be found in the market. Registered clinical studies are looking at the effects of CBD in insomnia (ACTRN12621001181897), epilepsy (NCT02987114), and hypertension (NCT05346562) using novel oral CBD products. Collectively, the results indicate that the lipid profile of a formulation substantially influences CBD delivery. Lipid-based oral formulations achieve higher systemic exposure (AUC/dose), whereas emulsion-based oral formulations demonstrate rapid systemic delivery (albeit not based on head-to-head studies). Depending on the PK requirements of a target indication (i.e., prolonged systemic exposure or rapid systemic delivery), an appropriate formulation could potentially be selected to best suit the patient’s needs.

## 4. Novel Routes of CBD Administration

As well as the many strategies being investigated to improve the oral bioavailability of CBD, researchers are investigating novel routes of CBD administration, some of which might be particularly relevant to specific indications. The PK profiles of different delivery strategies and administration sites (both preclinical and clinical) are reviewed below, and clinical developments in these delivery routes are highlighted. Figure 5 shows the PK indices for different routes of administration in various species, and the associated data are shown in Table 3.

**Table 3 pharmaceuticals-17-00244-t003:** PK parameters for different CBD formulations administered by non-oral routes in dogs, guinea pigs, horses, mice, rats, and humans. AUC, total drug exposure area under the curve; AUC/dose, dose-normalised area under the curve; CBD, cannabidiol; C_max_, maximum plasma concentration; C_max_/T_max_/dose, dose-normalised rate of systemic delivery; DBC, dimethyl-β-cyclodextrin; EtOH, ethyl alcohol; GML, glyceryl monolinoleate; MCT, medium-chain triglyceride; NP, nanoparticle; PEG, polyethylene glycol; PG, propylene glycol; PK, pharmacokinetics; T_max_, time to maximum concentration.

Route	Dose (mg/kg) *	Vehicle	Formulation Category	T_max_ (h)	C_max_ (ng/mL)	AUC (ng/mL × h)	C_max_ (ng/mL)/Dose (mg/kg)	AUC (ng/mL × h)/Dose (mg/kg)	C_max_ (ng/mL)/T_max_ (h)/Dose (mg/kg)	Ref
Rat										
Intravenous	0.2	PEG:saline:EtOH	Liquid	–	3596	104	17,980	518	–	[65]
Intravenous	2.5	Capryol + CremophorEL + TranscutolP + water	Liquid	0.08	1178	623	471	249	5654	[15]
Intravenous	4	PG	Liquid	0.08	2100	1380	525	345	6300	[7]
Intravenous	15.5	Saline	Liquid	0.5	5216	26,870	337	1734	673	[66]
Intravenous	4	PG	Liquid	0.08	872	674	218	168	2724	[14]
Intravenous	5	Saline	Liquid	–	–	1446	–	289	0.00	[22]
Intranasal	0.2	PEG	Liquid	0.35	35.4	152	177	758	506	[65]
Intranasal	0.2	PEG:saline:EtOH	Liquid	0.48	27.4	43.3	137	216	285	[65]
Intranasal	0.2	PEG:saline:EtOH: glycocholate	Liquid	0.47	30.2	41.8	151	209	322	[65]
Intranasal	0.2	PEG:saline:EtOH:DBC	Liquid	0.33	19.9	34.6	100	173	302	[65]
Intranasal	15.5	Saline	Liquid	0.5	2710	13,038	175	841	350	[66]
Intranasal	15.5	Nanoemulsion	Emulsion-based	0.5	2610	14,723	168	950	337	[66]
Inhalation	0.9	Sunflower oil	Liquid	5	217	132	1	8.58	0.11	[11]
Inhalation	1.7	Ethanol	Liquid	5	263	227	22	46.6	44	[11]
Inhalation	3.5	MCT	Liquid	5	567	513	10	29.2	4.83	[11]
Inhalation	6.7	PG	Liquid	5	888	950	241	147	48.2	[11]
Inhalation	13.9	PG	Liquid	60	2400	2030	155	134	30.9	[11]
Inhalation	10	PG	Liquid	0.5	220	466	173	146	2.88	[20]
Subcutaneous	10	PG	Lipid-based	7	8	85.8	133	142	26.5	[20]
Intramuscular	4.2	Nanocrystals	Nanotechnology	0.69	239	1459	57.0	348	82.6	[14]
Horse										
Intravenous	1	Chremophor + EtOH + saline	Liquid	0.08	1000	580.4	1000	580	12,000	[23]
Dog										
Intravenous	2.25	EtOH	Liquid	–	–	2706	–	1203	–	[67]
Intravenous	4.5	EtOH	Liquid	–	–	6095	–	1354	–	[67]
Intranasal	1.67	PEG:NaCl	Liquid	0.49	28	61.3	16.8	36.8	34.2	[26]
Transdermal	5.77	Cream	Lipid-based	10	74.3	11.7	12.9	2.03	1.29	[24]
Transdermal	11.5	Cream	Lipid-based	12	278	29.7	24.1	2.57	2.00	[24]
Mice										
Intravenous	10	Soybean oil + fat	Emulsion-based	0.17	2343	3191	234	319	1403	[28]
Intranasal	30	Suspension	Liquid	0.29	1205	2968	40.2	98.9	139	[29]
Guinea pigs										
Intravenous	1	PG	Liquid	0.03	269	175	269	175	8070	[65]
Transdermal	22.5	Gel (PG:water)	Solid/Resin	38.4	8.6	276	0.38	12.3	0.01	[65]
Transdermal	22.5	Gel (PG:water) + TranscutolHP	Solid/Resin	31.2	35.6	888	1.58	39.5	0.05	[65]
Human										
Intravenous	0.27	EtOH	Liquid	0.05	686	1000	2573	3751	51,450	[68]
Inhalation	0.26	Marijuana cigarettes	Smoke	0.05	110	291	430	1137	8594	[68]
Inhalation	0.13	DPI	Solid	0.06	18.78	7.66	141	57.5	2224	[43]
Inhalation	1.33	CBD dominant	Aerosol	0.10	171	151	128	113	1283	[46]
Inhalation	1.33	Pure CBD	Aerosol	0.10	105	73.9	78.5	55.4	785	[46]
Inhalation	1.33	CBD dominant	Aerosol	0.2	181.4	185.8	136	139	680	[55]
Inhalation	1.33	Pure CBD	Aerosol	0.2	104.6	108	78.5	81.0	392	[55]
Transdermal	1.33	Gefion GT4 tech	Emulsion-based	8	0.58	3.33	0.43	2.50	0.05	[69]

* Dose was normalised to a 75 kg human, ‘–’, values not reported.

### 4.1. Transdermal

Transdermal or topical administration of CBD is common, and clinical trials have shown that topical CBD significantly reduces pain in people with joint arthritis [70], peripheral neuropathy [71], and fragile-X syndrome [72] compared with placebo. Unfortunately, only a small number of studies have examined the PK parameters of CBD after transdermal administration. Transdermal administration of CBD is the slowest form of systemic delivery with a low AUC/dose (Figure 5).

A preclinical study conducted in dogs compared the PK profiles of oral microencapsulated oil beads, oral CBD-infused oil, and CBD-infused transdermal cream at doses of 10 or 20 mg/kg [24]. Acute application of CBD-infused transdermal cream led to detectable CBD that peaked at 10 h (10 mg/kg) or that was still increasing at 12 h (20 mg/kg), but with much lower plasma exposure than either of the oral formulations at equivalent doses. However, with repeated dosing for 6 weeks (twice per day), the CBD concentration was comparable between the (high-dose) transdermal and oral formulations. In hairless guinea pigs treated with a transdermal CBD gel, a steady-state plasma concentration was achieved at 16 h and was increased by 4-fold when administered with the permeability enhancer diethylene glycol monoethyl ether, with a T_max_ of 31 h [65]. This led to a 3-fold increase in systemic exposure and a 5-fold increase in the rate of systemic delivery.

According to a study in humans, the systemic CBD concentration after transdermal delivery is low. However, CBD was administered in combination with THC, which may have affected the results [69]. Nonetheless, using the Gefion GT4 transdermal delivery system, participants had detectable CBD within 2 h, with a T_max_ of 8 h. The maximal concentration of CBD reported in this study was 0.58 ng/mL, which is considerably lower than with other delivery methods. However, this was a single-dose study, and exposure levels are likely to increase with repeated application of CBD. Overall, the bioavailability of CBD was poor (Figure 4). Several studies have examined strategies to improve dermal delivery of CBD, but without accompanying systemic measurements of CBD or its metabolites. For example, studies have demonstrated that CBD permeation was enhanced by oleic acid [73] or chitosan and zinc oxide nanoparticles [74]. The treatment of dermatological conditions with CBD is an emerging research area [75], and novel skin-permeable CBD formulations may prove efficacious in this field, although this is yet to be confirmed in the literature.

In summary, while transdermal delivery of CBD may have clinical benefit [70,71], there is a paucity of PK data of transdermal CBD formulations in humans. Preclinical studies suggest that there is very slow permeation of CBD into systemic circulation, but twice daily application for 6 weeks leads to comparable CBD levels as with oral administration [24]. Thus, transdermal delivery of CBD is unlikely to deliver sufficient systemic levels of CBD for acute symptom management. Nevertheless, transdermal delivery may be useful for chronic disorders or where local delivery of CBD is required. Registered clinical studies are investigating the effects of topical CBD on vascular reactivity (NCT05456113), chemotherapy-induced pain (NCT05388058), painful osteoarthritis (ACTRN12621001512819), musculoskeletal pain (NCT05170451), and scar healing (NCT05650697). A permeation-enhanced CBD transdermal gel, Zygel™ [76], is currently in development for patients with fragile-X syndrome and 22q11.2 deletion syndrome (22q).

### 4.2. Inhalation

Because of the poor oral bioavailability of cannabinoids, their traditional route of delivery has been through inhalation. In rats, pulmonary delivery of CBD via a vaporiser (20 mg per 4 rats) led to detectable levels of CBD in serum (almost immediately) and brain (from 15 min) [20]. The peak serum concentration of CBD was similar between inhalational administration and oral administration (10 mg/kg) and higher than subcutaneous administration (10 mg/kg). The concentration of CBD in the brain was highest after oral delivery (pulmonary and subcutaneous delivery led to similar peak brain concentrations of CBD). Also in rats, CBD was more rapidly absorbed following inhalation (13.9 mg/kg) than by ingestion when administered with MCT oil by gavage (10 mg/kg; T_max_ 5 min vs. 2 h, respectively) with a 24-fold higher C_max_ and a 15-fold higher AUC [11]. The C_max_ and AUC of the metabolites 6-OH-CBD, 7-OH-CBD, and 7-COOH-CBD were also higher in the plasma after inhalation than after oral administration of CBD in male rats. Inhalation resulted in a slightly higher C_max_/T_max_/dose and a similar AUC/dose compared with oral administration, which resulted from significant absorption through the gastrointestinal tract [77].

A number of studies have examined the PK profile of CBD delivered through the lungs in humans. Ohlsson et al. (1986) compared inhalation and intravenous (IV) administration of deuterium-labelled CBD (19 mg) in healthy young males [68]. The estimated systematic bioavailability of inhaled CBD was approximately 31%. The mean clearance was around 1240 mL.min^−1^, and the mean distribution amount was 33 kg^−1^ (similar to what was previously observed with THC by the same investigators). In a study examining the urinary profile of CBD after administration, the urinary concentration of CBD was higher after oral administration than after CBD administered by vaporisation (100 mg in both cases), peaking at 5 h after ingestion and within 1 h after inhalation in healthy volunteers [78]. A follow-up study investigated the urinary PK profiles of three formulations of CBD (gelcap, pharmacy-grade syrup, and Epidiolex) and vaporised CBD, all 100 mg [64]. As before, the urinary CBD concentration was higher after oral administration, and the oral dose formulation affected the mean C_max_ (Epidiolex > capsule > syrup) but had little effect on T_max_ (~5 h with the oral formulation and 1.3 h with the vaporised formulation). The C_max_ of urinary metabolites of CBD was also highest with Epidiolex for 7-OH (Epidiolex > capsule > syrup > vaporised) and 7-CBD-COOH (Epidiolex > capsule = syrup > vaporised). A second publication from the same study reported the plasma profiles of CBD and its metabolites using the same cohorts [46]. Vaporised CBD delivered the highest plasma concentration of CBD, with the order of oral formulations being the same as that observed when measuring urinary CBD (Epidiolex > capsule > syrup). The 7-COOH-CBD concentration was 7-fold higher with oral administration than with vaporisation. Moreover, the comparison of inhalational (2.1 mg as a dry powder inhaler (DPI)) with oral (50 mg Epidiolex) CBD administration showed that the C_max_ was 71-fold higher with inhalational CBD, with a rapid onset (T_max_ 4 vs. 122 min, respectively) in healthy adults. The dose-adjusted AUC was also 9-fold higher with inhalational CBD than with oral CBD. Inhalational CBD resulted in a lower plasma concentration of 7-COOH-CBD than oral administration, with a metabolite-to-parent-drug ratio of 2.4 vs. 60, respectively. This suggests that there is reduced first-pass metabolism of the parent compound with inhalation. Devinsky and colleagues compared the PK profile of a DPI CBD formulation with the PK profile of Epidiolex in healthy volunteers. The authors found that DPI CBD had a 71-fold higher C_max_ than Epidiolex, while delivering 24-fold less CBD [43]. DPI CBD and Epidiolex had T_max_ values of 3.8 and 122 min, respectively. In humans, the C_max_/T_max_/dose for inhalation was ~50 times higher than in rodents as a larger amount of compound was absorbed from the respiratory tract [77], although the AUC/dose was lower with the use of specialised devices, suggesting significant device-related drug loss.

In summary, inhalation of CBD leads to rapid detection of CBD in the plasma and brain, which is likely to be particularly useful for indications requiring an immediate effect, such as seizures or panic attacks. Inhalation may lead to greater plasma exposure (AUC) than oral administration, although further direct comparison studies are needed. Inhalation may reduce the PK variability caused by gastrointestinal absorption and is subject to less first-pass metabolism. Regarding the potential clinical development of inhalational CBD products, investigators (Rapid Therapeutic Science Laboratories) have initiated two phase I trials with a metered-dose inhaler containing CBD in preparation for an Investigational New Drug application with the FDA.

### 4.3. Transmucosal

CBD users are commonly advised to put CBD oil under their tongue to allow absorption through the mucous membrane, which is suggested to have the potential advantage of faster delivery while avoiding first-pass metabolism. Sativex, the 1:1 THC:CBD medication used for the management of spasticity in multiple sclerosis, is administered via an oral cavity spray. However, a significant portion of the drug is washed from the mucosa and swallowed following its use, resulting in a similar PK profile to that of oral administration. Given the limited and very slow permeation of CBD across mucosal membranes, it is more likely that current transmucosal products are actually delivered orally.

In a preclinical study [79], a sublingual CBD–cyclodextrin complex and sublingual ethanolic CBD were compared with oral ethanolic CBD at a CBD dose of 250 µg/kg in rabbits. The AUC after sublingual administration of the CBD–cyclodextrin complex was similar to that of sublingual ethanolic CBD, but greater than that of oral ethanolic CBD in rabbits. Sublingual CBD was absorbed slowly, without a T_max_ during the 5 h study. A major limitation of the study was that the plasma concentration following oral administration was below the limit required for quantification, suggesting an issue with the absorption of the oral formulation. Another study conducted in pigs showed that CBD had a very low permeability rate over 8 h through the oral mucosa and that it accumulated within the oral mucosa [80]. In that study, CBD was detectable in the plasma, albeit appearing slowly and at a very low concentration. In the clinical setting, CBD is likely to be washed away by the saliva before mucosal permeation. In a study conducted in dogs, no difference was seen in any PK parameter between 1 mg/kg oral and transmucosal CBD administration [25]. The authors concluded that CBD was unlikely to be absorbed through the oral mucosa and was swallowed. One study in humans compared a sublingual wafer with an equivalent dose of oral oil solution in humans (both 50 mg CBD). The results showed no difference in the C_max_ or T_max_ (4 h) between the formulations [39].

Guy and Robson [58] compared sublingual, buccal, and oropharyngeal administration of liquid CBD spray in humans. Sublingual administration had a lower T_max_, whereas buccal administration had a higher C_max_ [58]. In another study, healthy volunteers [39] were administered a sublingual wafer (25 and 50 mg) or an oil-based solution to compare their CBD PK profiles. The porous and amorphous WaferiX™ matrix containing active CBD is designed to rapidly collapse and release CBD nanoparticles within 1 min of contact with the saliva in the sublingual space. When comparing the two wafer doses with eight sprays of Sativex and an MCT oil, there were no statistically significant differences between the AUC0–24 h of CBD after administration of MCT oil or wafer compared with Sativex.

Some studies have investigated various mechanisms to improve transmucosal permeation, without measuring PK variables. Two studies by Tabboon and colleagues [81,82] showed that diethylene glycol monoethyl (DEGEE) and PEG or their combination, or DEGEE with MCTs, enhanced CBD permeation through the oesophageal mucosa in pigs.

In summary, because of the lipophilic nature of CBD, transmucosal absorption is slow with a low permeability rate. Thus, the PK profile of CBD with oral mucosal delivery tends to be similar to the PK profile with oral ingestion (Figure 1). The AUC/dose of solidified formulations, such as wafer, is high; however, similar exposure and performance can be obtained with oral administration. It is possible that future studies will establish novel ways to improve oral permeation, but no data currently exist to suggest the superiority of this delivery route.

### 4.4. Intranasal

Intranasal (IN) drug administration is a potential route of delivery for neurological conditions [83], where CBD has most clinical evidence for efficacy [1]. Preclinical studies have shown that intranasal delivery of CBD improves symptoms in several behavioural tests of posttraumatic stress disorder [29] and produces long-lasting antinociceptive effects in neuropathic pain, with a faster onset than oral delivery [84].

PK studies have shown that CBD (200 μg/kg) is absorbed intranasally within 10 min, with a bioavailability of 34–46% in rats [65]. PEG 400-A leads to an increase of approximately 3.5-fold in AUC compared with the intranasal PEG 400-ES formulation. In this study, bioavailability did not increase with the addition of enhancers. When delivered at the same dose with the same formulation (PEG 400-ES), the AUC of CBD was around 2-fold higher with intravenous administration than with intranasal administration [65]. Ahmed and colleagues [66] also showed that the plasma AUC of intranasal CBD delivery (3.1 mg) was approximately half that of intravenous CBD delivery. However, the brain AUC was around 2-fold higher after intranasal administration of a CBD nanoemulsion formulation compared with intravenous or intranasal delivery of non-formulated CBD. This indicates that CBD formulated as a nanoemulsion is potentially able to pass through the olfactory epithelium, thereby bypassing the blood–brain barrier and increasing the CBD concentration in the brain. In dogs, CBD administered intranasally (20 mg) resulted in a C_max_ ~66% and an AUC ~22% of those of oral administration (100 mg) [26]. The T_max_ with intranasal CBD ranged between 20 and 30 min among studies [26,65,66].

In summary, intranasal administration of CBD appears to be a fast and effective delivery method, and brain exposure is enhanced with an optimised nanoemulsion formulation. Therefore, this may be an attractive route of delivery for neurological conditions. NobrXiol™ contains pharmaceutical-grade CBD in an intranasal spray powder formulation. It is being developed for the management of epilepsy [85], but it is not yet in clinical trials.

### 4.5. Intravenous

Although not a common route of administration, a limited number of preclinical studies have shown that intravenous administration of CBD is effective for the treatment of certain conditions, such as newborn brain damage after oxygen starvation [86], ischemia/reperfusion liver injury [87], and depression [28]. Ohlsson and colleagues [68] first described the PK profile of deuterium-labelled CBD (20 mg) in humans, showing that the AUC was 4-fold higher with intravenous administration than when administered by smoking/inhalation at a similar dose (19.2 mg). In another study, compared with intramuscular (5 mg/kg) and oral (15 mg/kg) administration, CBD administered intravenously (4 mg/kg) had a higher C_max_ and a similar half-life in rats [14]. The AUC of intravenous CBD was similar to that with oral delivery. In another study conducted in horses, the AUC of intravenous CBD (1 mg/kg) was slightly lower than that of two oral CBD formulations (both 10 mg/kg) [23]. In dogs, intravenous administration of CBD (45 or 90 mg) led to a dose-proportional increase in AUC, with a terminal half-life of approximately 9 h [67]. In the same study, oral CBD (180 mg) did not lead to detectable plasma CBD.

In a direct comparison of efficacy in mice, intravenous CBD (10 mg/kg) had a similar antidepressant effect to 100 mg/kg oral CBD when measured 1 h after drug administration [28]. According to acute PK measurements, mice treated with intravenous CBD had a plasma concentration of ~370 ng/mL compared with a plasma concentration of ~50 ng/mL in mice treated with oral CBD. However, the animals were treated weekly with the respective doses of intravenous or oral CBD, so it is unclear how much of the antidepressant effect was due to chronic dosing and how much was due to acute dosing.

### 4.6. Intramuscular

Intramuscular delivery of CBD is not common. However, one preclinical study showed that intramuscular injection of CBD (5 mg/mL resulting in administration of 10 µg CBD) reduced mechanical sensitivity in rats, suggesting possible benefits in the context of chronic muscle pain [88]. A recent study showed that intramuscular delivery of a CBD nanocrystal formulation (5 mg/kg) led to enhanced plasma CBD exposure (AUC) when compared with intravenous administration (4 mg/kg) or oral administration of the same CBD nanocrystal or CBD dissolved in oil (both 15 mg/kg) [14]. Intramuscular administration also had a higher C_max_ and a faster onset (T_max_ ~40 min) than oral administration (T_max_ 75–240 min depending on the formulation). Further studies are warranted on intramuscular administration as a potential method to achieve fast and effective CBD delivery.

### 4.7. Intrarectal

Intrarectal administration is a useful delivery mechanism for certain patients, such as those who cannot swallow or are unconscious, or for indications requiring improved bioavailability or local delivery. Moqejwa and colleagues [89] developed novel CBD-loaded transferosomes for intrarectal delivery with improved permeation of colorectal membranes compared to CBD alone, without affecting membrane integrity. Further PK or efficacy assessments were not conducted for these formulations. However, intrarectal administration of CBD (100 mg suppository after defecation) in dogs did not lead to detectable plasma CBD, suggesting that intrarectal administration may not be an appropriate mechanism for systemic CBD delivery [26]. Future studies are needed to evaluate local administration for indications such as inflammatory bowel disorders, for which intrarectal administration may be appropriate.

### 4.8. Subcutaneous

Only one study to date has examined the PK profile of CBD after subcutaneous administration. The concentrations of CBD in the serum and brain peaked at around 1 h after subcutaneous delivery of CBD (10 mg/kg) in rats [20]. Compared with oral delivery of the same dose, peak serum and brain concentrations were 3–4-fold lower with subcutaneous delivery, despite appearing to have a longer half-life; however, the AUC and half-life were not reported. Subcutaneous administration showed two CBD concentration peaks, possibly indicative of a two-compartment (central and peripheral) PK model.

At least two subcutaneous CBD products are currently in development. CRD-38 is a novel subcutaneously administered formulation of CBD that is intended for use in heart failure [90]. Valeritas has also developed a proprietary h-Patch™ CBD subcutaneous delivery system, although the status of this development programme is unclear [91].

## 5. Tissue Distribution of CBD

A limited number of studies have investigated the tissue distribution of CBD and whether CBD distribution is affected by the CBD formulation or route of administration.

In a guinea pig model, CBD was detectable in joint tissues (articular cartilage and infrapatellar fat pad of the stifle joints) 24 h after oral administration of 25 or 50 mg/kg CBD in almond oil [92]. The concentration of CBD in the joint tissues was dose-dependent, with the CBD concentration being higher in the infrapatellar fat pad (~260 ng/g; 50 mg/kg) than in the articular cartilage (~80 ng/g; 50 mg/kg). CBD was also detected in the synovial fluid of horses administered oral CBD acutely (3 mg/kg in sunflower oil) or for 6 weeks (0.5 or 1.5 mg/kg) [93]. CBD was more consistently detectable after 5 weeks of twice-daily dosing (~1–6 ng/mL) compared to 12 h after the first acute dose, suggesting that CBD accumulates in the synovial fluid.

In rats, CBD and its metabolites (6-OH-CBD and 7-OH-CBD) were detected in the whole brain within 5 (CBD) or 30 (metabolites) min after oral dosing (10 mg/kg CBD in MCT). The T_max_ was around 2 h, and the maximal CBD brain exposure was around 100 ng/mL [11]. Oral administration of CBD (10 mg/kg in sunflower oil) resulted in a peak brain concentration of around 200 ng/g at 2 h after administration [20]. In the same study, the concentration of CBD in the brain was measured after subcutaneous dosing of CBD (10 mg/kg), peaking at around 45 ng/g at 1 h. After acute pulmonary administration of CBD (20 mg vaporised per four rats), the peak concentration was around 75 ng/g at 15 min [20].

Different CBD formulations may lead to different CBD brain concentrations among brain regions. Microcapsule oral administration of CBD (5 mg/kg in MCT oil) with deoxycholic acid, a permeation-modifying bile acid, showed an earlier and slightly higher (although not statistically significant) CBD concentration in the whole brain compared with two other preparations [27]. A recent study found that oral administration of different lipid-based CBD formulations resulted in varied brain CBD exposures [10]. All oral formulations (lipid-free, sesame oil, coconut oil, and rapeseed oil) showed peak brain exposure at around 1 h after administration (12 mg/kg), with the highest CBD concentration detected in the olfactory bulb, occipital lobe, and striatum. Administration of CBD in lipid-free vehicle resulted in the highest whole-brain CBD exposure (AUC: 3270 ± 1890 ng/g × h), followed by the sesame oil, coconut oil, and rapeseed oil formulations. The relative distribution of CBD across brain regions was formulation-dependent. For example, the CBD concentration in the striatum (motor and reward systems) was considerably higher with the sesame oil, coconut oil, and rapeseed oil formulations. After intranasal delivery, the concentration of CBD in the brain was greater with an optimised nanoemulsion of CBD than with either intravenous or intranasal delivery of non-formulated CBD [65].

Another study measured the accumulation of CBD in the gastrocnemius muscle, liver, and adipose tissue in adult rats following oral gavage at increasing doses (0, 30, 115, and 230 mg/kg/day in MCT oil) [94]. After dosing for 28 days, CBD was around 100-fold higher in the adipose tissue than in the muscle or liver tissue. For example, at the maximum dose of 230 mg/kg/day, the CBD concentration was 1.15 mg/kg in the liver, 0.93 mg/kg in muscle tissue, and 109 mg/kg in adipose tissue. For the same relative CBD dose, females consistently showed higher concentrations in the muscle and liver. Repeated pulmonary dosing of CBD over 14 days (6.7 or 13.9 mg/kg) led to quantifiable levels of CBD in the brain measured 24 h after the last dose, while CBD metabolites were only measurable in the brain tissue following the highest inhaled dose (13.9 mg/kg) [11].

In summary, a limited number of studies have shown that after acute oral CBD dosing, CBD is detectable in the joints and synovial fluid, as well as in the brain, albeit with regional heterogeneity in the brain. This distribution profile may be dependent on the CBD formulation. CBD is also detectable in the brain when administered by inhalation (fast onset), subcutaneously, and intranasally. Data suggest that the formulation and route of delivery of CBD may be optimised for delivery to specific brain regions. Repeated CBD dosing leads to a higher detectable concentration of CBD in the adipose tissue than in the liver and muscle tissue and is also detectable in the synovial fluid (oral administration) and brain (inhalational administration).

## 6. Does an Improved CBD PK Profile Lead to Better Efficacy?

Although some products have improved the plasma exposure to CBD, it remains to be established whether this increased plasma exposure equates to superior therapeutic efficacy. Some studies have attempted to answer this question, and others have evaluated whether particular routes of administration are better suited to some conditions.

In one study in rats, a CBD emulsion demonstrated higher absolute oral bioavailability than an oil solution, with greater lymphatic absorption of CBD [15]. The same study also showed that the CBD emulsion was therapeutically effective in a model of rheumatoid arthritis. Unfortunately, there was no control arm with which to compare the oil CBD solution in the rheumatoid arthritis model, so it was not possible to assess whether the enhanced bioavailability of the emulsion led to superior efficacy. In another study, the administration of nasal inclusion complex temperature-sensitive hydrogels resulted in a higher brain tissue distribution and improved PK compared to oral administration in mice [29]. The nasal formulation was effective in a model of posttraumatic stress disorder, but no direct comparison with the efficacy of oral CBD was made in the same model.

In humans, the plasma concentration of CBD over time was higher in volunteers treated with a corn oil formulation than in those treated with a powdered formulation (both 150 mg). However, neither of the formulations affected a facial emotion recognition task [51]. Hobbs et al. [41] showed that in healthy volunteers, a water-soluble CBD powder demonstrated greater bioavailability than a lipid-soluble formulation (both 30 mg). However, there was no difference in tumour necrosis factor or interleukin-10 production in lipopolysaccharide-stimulated or non-stimulated peripheral blood mononuclear cells collected from volunteers 90 min after CBD exposure.

Other studies have concluded that certain CBD formulations have better efficacy, without concurrent PK studies. For example, CBD complexes with cyclodextrins were more cytotoxic to cancer lines in vitro than CBD dissolved in dimethyl sulfoxide, indicating their potential applicability in the field of oncology [95]. Similarly, in another study, poly-ε-caprolactone microspheres of CBD were slightly more effective at killing breast cancer cells in vitro [96]. Small lipid nanocapsules of CBD were also better at killing brain cancer cells in vitro [97]. In the pain setting, a nanostructured lipid carrier of CBD delivered intranasally produced a faster and longer-lasting antinociceptive effect in animals with chemotherapy-induced neuropathic pain than oral administration or nasal administration (both 5 mg/kg) of a CBD solution [84]. CBD-loaded poly lactic-co-glycolic acid nanoparticles were more effective than CBD at reducing inflammation in chondrocytes in an in vitro model of osteoarthritis [98].

A small number of preclinical studies have compared how the route of CBD administration affects its therapeutic profile. Schicho and colleagues compared the effects of CBD administered intraperitoneally (10 mg/kg), orally (20 mg/kg), or intrarectally (20 mg/kg) in a mouse model of colitis [99]. The authors found that intraperitoneal or intrarectal administration of CBD significantly improved the colitis score index and decreased myeloperoxidase activity, while oral dosing did not. Unfortunately, the plasma concentration of CBD was not measured, so it was not possible to determine whether the oral CBD product achieved a similar plasma concentration to other routes. In mice, repeated dosing with either 10 mg/kg CBD administered intravenously or 100 mg/kg CBD administered orally (but not 10 mg/kg administered orally) exerted antidepressive behavioural effects in the forced swim test in chronically stressed animals, but some of the molecular endpoints (synaptophysin and IBA1) were more consistently modified by intravenous dosing [28]. In a mouse model of brain cancer, local administration of CBD-loaded poly-ε-caprolactone microparticles (containing 6.7 mg CBD, given every 5 days) reduced tumour growth to the same level as daily local administration of the same amount of CBD (15 mg/kg for 22 days) [96]. In addition, subcutaneous and intranasal administration of CBD (both 6.7 mg/kg for 26 days) reduced the symptoms of multiple sclerosis in mice [100].

In conclusion, while some studies have demonstrated a superiority of novel CBD formulations in PK, this has not yet been linked to a greater therapeutic profile, or indeed to a reduced side effect profile. Similarly, some products have shown therapeutic superiority over non-formulated CBD, but without PK measurements to establish whether this is related to differences in the onset, peak, or duration of plasma/tissue CBD exposure. Direct comparisons of various routes of administration (with different associated PK profiles) have also not yet been linked to greater efficacy for particular indications. This is one area that would greatly benefit from further research.

## 7. Conclusions

This review of the published literature shows that research and commercial development are ongoing to improve the delivery, bioavailability, and variability of CBD. Maximising CBD delivery may help to improve therapeutic efficacy while reducing its side effect profile and drug–drug interaction risks, which is the ultimate aim for the patient. There are limited but promising data on novel delivery routes such as inhalation, intramuscular, and subcutaneous. However, evidence is limited regarding whether changes in CBD PK profiles equate to superior therapeutic efficacy across indications and whether specific CBD products might be suited to particular indications. Very limited data show that CBD is generally well distributed across tissues and that some CBD products enable increased delivery of CBD to different brain regions. Unique tissue distribution profiles associated with different products may be better suited to some indications than others. However, again, there is a paucity of data testing this. Thus, there is still much work to be done to optimise CBD products, and head-to-head comparisons should be performed across various indications.

## Figures and Tables

**Figure 1 pharmaceuticals-17-00244-f001:**
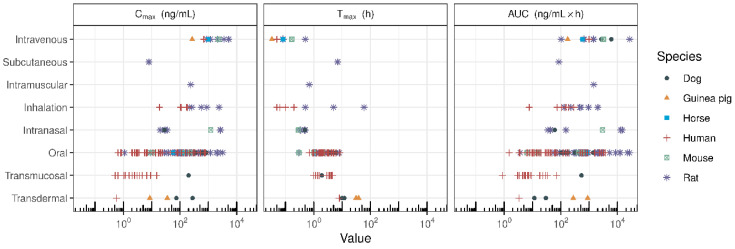
Maximum plasma concentration (C_max_), time to maximum plasma concentration (T_max_), and area under the curve (AUC) for cannabidiol administered by various routes in different species. The raw data used in Figure 1 are listed in Table 1, Table 2 and Table 3.

**Figure 2 pharmaceuticals-17-00244-f002:**
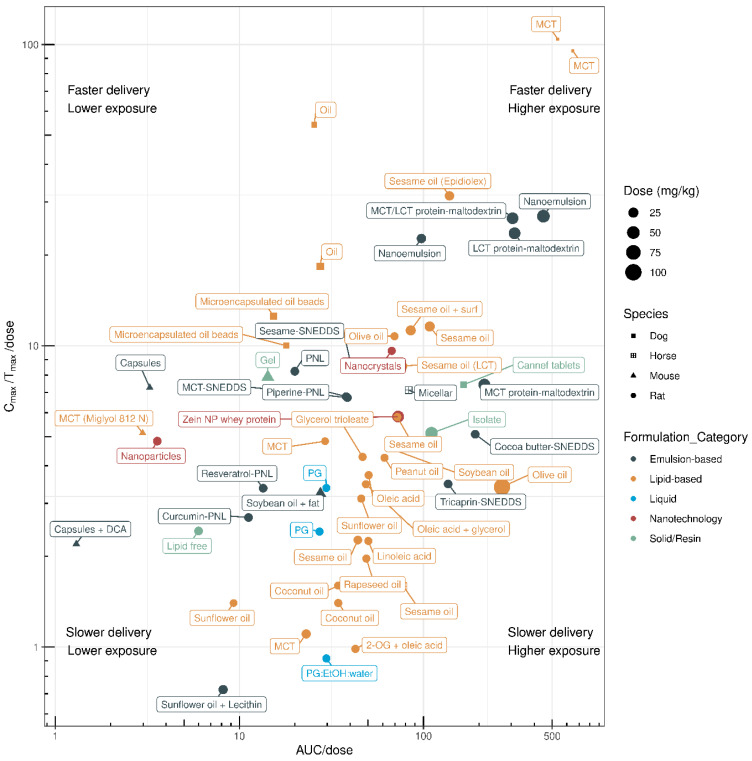
Oral delivery efficiency of various cannabidiol formulations in different species based on the PK indices. A dose-normalised systemic delivery rate (C_max_/T_max_/dose) and dose-normalised systemic exposure (AUC/dose) are used to evaluate cannabidiol-based formulation performance. The use of C_max_/T_max_/dose and AUC/dose enables the selectin of CBD formulation with desired rate of systemic delivery and overall exposure. The labels describe the vehicle used for cannabidiol formulation. The PK data of the CBD formulations in different species are listed in Table 1.

**Figure 3 pharmaceuticals-17-00244-f003:**
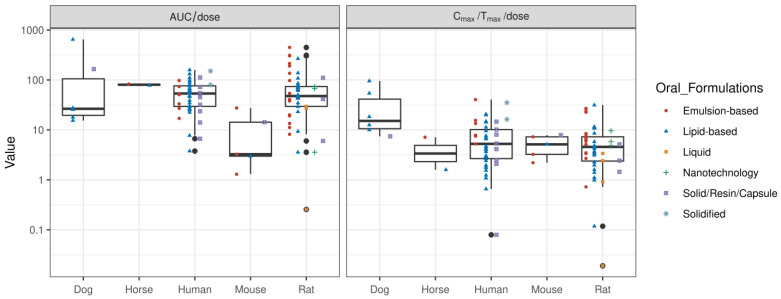
Box plot of overall dose-normalised systemic exposure (AUC/dose) and dose-normalised rate of systemic delivery (C_max_/T_max_/dose) for various oral formulations of CBD across different species.

**Figure 4 pharmaceuticals-17-00244-f004:**
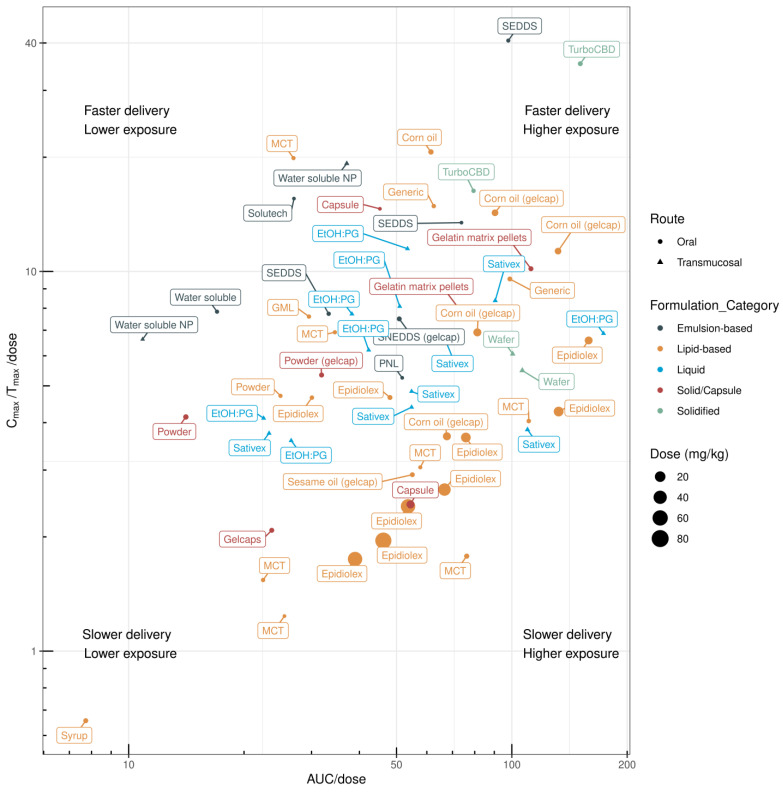
Oral delivery efficiency of different CBD formulations in humans. The efficiencies were compared by deriving the pharmacokinetic indices, i.e., dose (mg/kg)-normalised area under the curve (AUC/dose) and dose-normalised rate of systemic delivery (C_max_/T_max_/dose). The data are provided in Table 2.

**Figure 5 pharmaceuticals-17-00244-f005:**
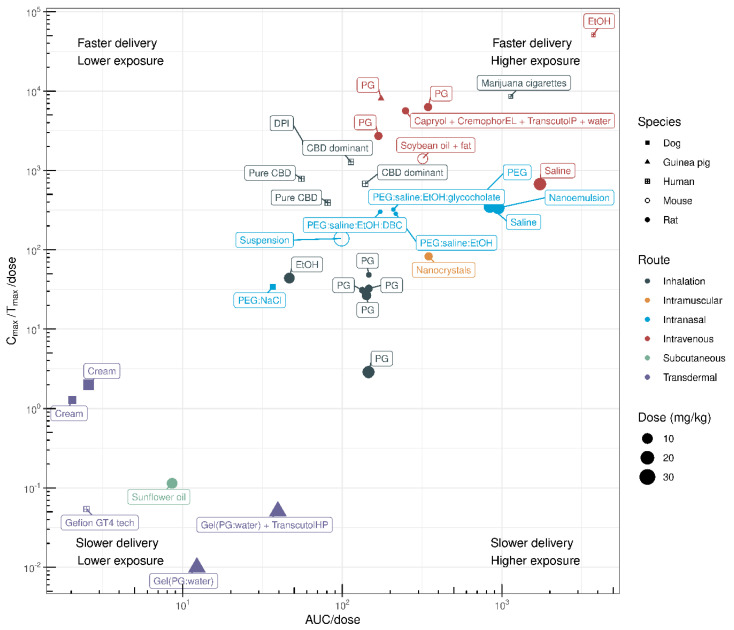
Multi-route delivery efficiency of CBD in different species based on PK indices. The systemic exposure of CBD is derived using dose-normalised area under the curve (AUC/dose) and the rate of systemic delivery is estimated using dose-normalised C_max_/T_max_ (C_max_/T_max_/dose). C_max_ (ng/mL) is the maximum plasma concentration; and T_max_ (h) is the time to maximum concentration. The data are provided in Table 3.

## Data Availability

Data sharing is not applicable.

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
