# Peer review of "Strategies to Improve Cannabidiol Bioavailability and Drug Delivery"

_pharmaceuticals, 2024, doi:10.3390/ph17020244_

Round 1
Reviewer 1 Report
Comments and Suggestions for Authors
The primary objective of the manuscript by O'Sullivan, Jensen, Kolli, Nikolajsen, Ziegler, Bruun, and Hoeng was to evaluate the preclinical and clinical pharmacokinetics of various formulations that include cannabidiol, from oral formulations to new methodologies to improve the bioavailability of this naturally active compound. The secondary objective was to examine the pharmacokinetics of cannabidiol administered by various routes to improve systemic exposure. The literature search included mainly recent papers (2023-2019), but also older studies dating back to 1988 (reference 78), which underscore the interest in this critical topic that nevertheless still has critical issues. Particularly, the tissue distribution of cannabidiol and whether distribution is affected by the cannabidiol formulation or route of administration are the main shortcomings outlined by the authors.
The decision of this reviewer is "Accepted after minor review," for the comments below
Typographical errors:
· Title of paragraphs in block letters or not
· “3. STRATEGIES TO IMPROVE THE ORAL BIOAVAILABILITY OF CBD (PRECLINICAL STUDIES)”, however the section refers to clinical studies
Author Response
The primary objective of the manuscript by O'Sullivan, Jensen, Kolli, Nikolajsen, Ziegler, Bruun, and Hoeng was to evaluate the preclinical and clinical pharmacokinetics of various formulations that include cannabidiol, from oral formulations to new methodologies to improve the bioavailability of this naturally active compound. The secondary objective was to examine the pharmacokinetics of cannabidiol administered by various routes to improve systemic exposure. The literature search included mainly recent papers (2023-2019), but also older studies dating back to 1988 (reference 78), which underscore the interest in this critical topic that nevertheless still has critical issues. Particularly, the tissue distribution of cannabidiol and whether distribution is affected by the cannabidiol formulation or route of administration are the main shortcomings outlined by the authors.
The decision of this reviewer is "Accepted after minor review," for the comments below
Typographical errors:
- Title of paragraphs in block letters or not
**thank you for this observation – all titles are not in lower caps only.
- “3. STRATEGIES TO IMPROVE THE ORAL BIOAVAILABILITY OF CBD (PRECLINICAL STUDIES)”, however the section refers to clinical studies
**thank you for this observation –this has been corrected.
Reviewer 2 Report
Comments and Suggestions for Authors
Please correct references according to journal guidelines.
Please add comments in the figure captions in order to improve the description of the results.
Tables should be moved and included soon after their description in the text.
Comments on the Quality of English Language
Minor corrections
Reviewer 3 Report
Comments and Suggestions for Authors
pharmaceuticals-2844253
Strategies to improve cannabidiol (CBD) bioavailability and drug delivery
The manuscript reviewed the literature on CBD to identify novel oral products and delivery strategies for CBD. The manuscript was well prepared and could contribute considerably to the field. However, please consider the following comments to improve this manuscript before publication.
1. The abbreviation should be removed from the title.
2. This review can be modified to become a systematic review following the PRISMA guidelines. I suggest that the authors consider modifying the manuscript by including a method section and describing how the data were collected and analyzed and how the graphs were constructed.
3. Raw data of the figure 1 should be mentioned/ included.
4. Conclusion: The authors should conclude which administration route and drug delivery systems are promising.
Comments on the Quality of English LanguageMinor editing of English language required
Reviewer 4 Report
Comments and Suggestions for Authors
I trust this message finds you well. I have had the opportunity to review your review with great interest since I have been working on this plant ( Cannabis Sativa) for so many years.
The paper is well-written, and your exploration of the literature is thorough. However, it appears that the methodology may not fully adhere to the expected standards. In particular, there is a need for more clarity and explicit details regarding the methodology employed in your review. I appreciate the significance of your research and believe that addressing this concern will further strengthen the overall quality of your work.
also Line 41 a reference is needed
Please improve the quality of figure 1, the same for figure 2
Line 71 no need to be capitalised
Round 2
Reviewer 3 Report
Comments and Suggestions for Authors
The manuscript was appropriately revised and can be accepted.
Reviewer 4 Report
Comments and Suggestions for Authors
authors have made the necessary revision